# Productive Performance, Physiological Variables, and Carcass Quality of Finishing Pigs Supplemented with Ferulic Acid and Grape Pomace under Heat Stress Conditions

**DOI:** 10.3390/ani13142396

**Published:** 2023-07-24

**Authors:** María A. Ospina-Romero, Leslie S. Medrano-Vázquez, Araceli Pinelli-Saavedra, Esther Sánchez-Villalba, Martín Valenzuela-Melendres, Miguel Ángel Martínez-Téllez, Miguel Ángel Barrera-Silva, Humberto González-Ríos

**Affiliations:** 1Centro de Investigación en Alimentación y Desarrollo, A.C. Carretera Gustavo Enrique Astiazarán Rosas, No. 46, Col. La Victoria, Hermosillo 83304, Mexico; mospina221@estudiantes.ciad.mx (M.A.O.-R.); lmedrano121@estudiantes.ciad.mx (L.S.M.-V.); pinelli@ciad.mx (A.P.-S.); martin@ciad.mx (M.V.-M.); norawa@ciad.mx (M.Á.M.-T.); 2Departamento de Agricultura y Ganadería, Universidad de Sonora, Carretera a Bahía de Kino km 21, Hermosillo 83000, Mexico; esther.sanchez@unison.mx

**Keywords:** feedlot performance, grape pomace, ferulic acid, by-products, phenolic compounds, pork production

## Abstract

**Simple Summary:**

Phenolic compounds, and sources rich in these secondary metabolites, have received great interest in the monogastric nutrition area and have been considered natural alternatives to the use of synthetic growth promoters in intensive swine production. The present study evaluated physiological variables, productive performance, and carcass quality of finishing pigs supplemented with ferulic acid (FA) and grape pomace meal (GPM) for 31 days prior to slaughter under heat stress conditions. The inclusion of FA and GPM in the diet did not affect any of the physiological variables evaluated. On the contrary, GPM increased the feed intake (FI), while the FA addition modified hot and cold carcass yields. Likewise, GPM decreased the marbling degree of the carcasses. The results indicate that the combined supplementation of FA and GPM is favorable for carcass characteristics.

**Abstract:**

The effect of individual and combined supplementation of FA and GPM on physiological variables, productive performance, and carcass characteristics of finishing pigs under heat stress conditions were investigated. Forty Yorkshire × Duroc pigs (80.23 kg) were individually housed and randomly distributed into 4 groups under a 2 × 2 factorial arrangement (*n* = 10): Control (basal diet, BD); FA, BD + 25 mg FA; GPM, BD with 2.5% GPM; and MIX, BD with 25 mg FA and 2.5% GPM. Additives were supplemented for 31 days. The inclusion of FA or GPM did not modify rectal temperature and respiratory rate. There was an effect of the interaction on FI, which increased when only GPM was supplemented, with respect to Control and MIX (*p* < 0.05). Average daily gain (ADG) and feed conversion (FC) were not affected by treatments (*p* > 0.05). The inclusion of FA improved hot and cold carcass weight, while the addition of GPM decreased the marbling (*p* < 0.05) and tended to increase loin area (*p* < 0.10). GPM increased liver weight (*p* < 0.05). The addition of GPM and FA can improve some carcass characteristics under heat stress conditions. It is necessary to continue investigating different levels of inclusion of GPM and FA in finishing pigs’ diets.

## 1. Introduction

Environmental stressors such as high temperatures and relative humidity represent one of the main challenges for swine production systems worldwide [1]. Heat stress is an environmental issue that negatively affects animal welfare indicators and productive performance, and generates economic losses for producers [2]. When the environmental temperature exceeds the thermoneutral zone in finishing pigs (18 to 25 °C) heat stress is induced, the animal prioritizes dietary energy for thermoregulatory function, the productive performance is negatively affected, and an imbalance in the health status of the animal may occur [1]. This is considered a limiting factor in the meat supply to the population [2].

On the other hand, to improve the growth performance of pigs during their finishing stage, intensive systems have resorted to various synthetic compounds such as β-adrenergics and antibiotics, with which it has been possible to improve parameters such as average daily gain (ADG), feed intake (FI), feed conversion (FC), and carcass quality [3]. However, their use has been restricted and banned in several countries in response to the risks they pose to public health and changes in consumer purchasing trends [4]. In this sense, the addition of secondary plant metabolites (phytochemicals) in monogastric diets has been evaluated as a natural alternative to the use of synthetic growth promoters, seeking to obtain simultaneous benefits in animals [5,6].

Previous studies in pigs have demonstrated that phenolic compounds (PCs) such as ferulic acid (FA) and those present in various agro-industrial by-products such as grape pomace (GP) exert multiple biological activities (antioxidant, growth promoter, antimicrobial, immunomodulatory, etc.) and numerous action mechanisms [6,7]. The FA has been considered a potential growth modulator in pigs. So far, different doses (12, 15, 25, and 100 ppm) of FA have been evaluated for periods between 27 to 35 days prior to slaughter and have reported an improvement in productive performance, transition in the type of muscle fibers, increased activity of antioxidant enzymes and reduction of malondialdehyde levels [6,8]. 

Regarding GP, it is a rich source of PCs with high antioxidant capacity, so it has great potential to be included as a dietary additive. Recently, it has been reported that some of its PCs also have potential as growth modulators, since it was shown that they can promote muscle fibers transition and regulate lipid metabolism [9,10]. In addition, to enhance the beneficial effects of individual PCs, dietary supplementation of mixtures of extracts has been studied, looking for possible synergies between them, and to obtain simultaneous benefits in the animal [11]. This may represent a strategy to counteract pre-slaughter stress in pigs and guarantee an improvement in productivity without detriment to animal welfare and meat quality.

To our knowledge, combined supplementation of FA and GP has not been reported against stressful stimuli and as an animal growth promoter [12]. Therefore, the aim of this study was to evaluate the effect of individual and combined supplementation of FA and grape pomace meal (GPM) on productive performance, physiological variables, blood metabolites, and carcass characteristics in finishing pigs exposed to heat stress conditions.

## 2. Materials and Methods

### 2.1. Preparation of Grape Pomace Meal and Nutritional Components

The GP from the Tempranillo cultivar with 48 h after pressing was supplied by an industrial wine production company in Ensenada, Baja California, and was dried in a convection oven (ENVIRO-PAK, model Micro-Pak, series MP500, Clackamas, OR, USA) at 60 °C during 6 h and 20 min to obtain a product with a moisture content of less than 10%. Dry GP was milled to a particle size of 1 mm in Pulvex 200 (Molinos Pulvex, S.A. de C.V. Ciudad de México, Mexico) equipment to obtain the GPM, which was packed in vacuum-sealed plastic bags, protected from light, and kept to 4 °C until use. The GPM was incorporated in the conventional feed at a proportion of 2.5%. The inclusion percentage was selected according to the total phenol content estimated in GPM and this did not exceed 1500 mg/kg feed of phenolic compounds per animal/day [13].

Moisture content (930.15), ash (942.05), protein (984.13), ethereal extract (920.39), and crude fiber (962.09) were determined in triplicate according to the methods described by the Association of Official Analytic Chemist (AOAC, 1990) [14]. Neutral detergent and acid detergent fiber were also quantified [15]. The nutrient content of GPM was 7.88% moisture, 13.4% crude protein, 7.61% fat, 24.12% crude fiber, 28.43% acid detergent fiber (ADF), 40.48% Neutral detergent fiber (NDF), 7.09% ash and 2.56 Mcal/kg of metabolizable energy.

### 2.2. Antioxidant Capacity, Phenolic Compounds Content and Chemical Characterization of GPM

#### Obtaining Extracts and Measurement of Phenolic Content in GPM

Methanolic extracts of GPM were prepared [16] and used to quantify the total content of phenolic compounds, flavonoids, and antioxidant capacity. One g of GPM was homogenized in 10 mL of 80% methanol, subsequently sonicated for 30 min, and centrifuged at 9400 rpm for 15 min. The supernatant was collected and filtered on Whatman N°1 paper and the residues were washed twice more with 80% methanol and stored at −20 °C. The total phenolic compound content was estimated using the Folin Ciocalteu technique and gallic acid standard calibration curve. The absorbance was read at 765 nm using a FLUOstar Omega spectrophotometer (BMG Labtech, Durham, NC, USA), and was expressed as mg gallic acid equivalents (GAE)/g dw. Flavonoid content was determined by spectrophotometric methods [17]. Absorbance was measured at 512 nm on the same spectrophotometer (BMG Labtech) and was expressed as mg catechin equivalents (CE)/g dw. Antioxidant capacity was determined by FRAP (ferric reducing antioxidant power), TEAC (trolox equivalent capacity), and DPPH methods and expressed in µmol of Trolox equivalents. All analyses were performed in triplicate.

Qualitative characterization of phenolic compounds was carried out by thin-layer chromatography using Nano-SIL NH2 UV254 (Macherey-Nagel 4 × 10 cm) aminated silica gel plates [18]. A mixture of butanol: methanol: water: formic acid (50:25:20:1) was used as eluent phase and the phenolic compound standards: ferulic acid (FA), gallic acid (GA), chlorogenic acid (CGA), caffeic acid (CA), catechin (CAT), epicatechin (EC), and resveratrol (RES), were placed in different lanes together with the grape pomace samples (12 µL). The chromatoplate was visualized with UV light (254 nm and 365 nm) and subsequently developed (p-anisaldehyde, sulfuric acid). 

### 2.3. Animal Feeding Trial

All procedures involving handling and animal slaughter were conducted according to official Mexican standards [19,20] and this study was approved by the ethics committee of the Universidad de Sonora. With respect to the supplemented additives, the GPM described in Section 2.1 was used, while the FA was food grade with a high degree of purity (95%) and obtained from Laboratorios Minkab S.A de C.V. (Guadalajara, Jalisco, Mexico).

#### 2.3.1. Animals and Treatments

The study was carried out in the pig experimental unit of the Agriculture and Livestock Department (ALD) of the Universidad de Sonora, (UNISON) Hermosillo, Northwestern Mexico, during the months of May and June 2022. The average temperature and relative humidity during the study were 29.9 ± 3.01 °C and 38.6 ± 8.3%, respectively. Forty male pigs from commercial Duroc × Yorkshire crossbreed with an initial live weight of 80.2 ± 4.6 kg were used. The animals were individually housed (0.6 × 2.0 m) in open buildings representing local commercial facilities for pigs during the grow-finish phase. Pigs were given ad libitum access to nipple-type drinkers and feeding troughs. 

The feeding test was carried out during the finishing stage for a period of 31 days, and 10 pens were randomly assigned to one of the following treatments under a completely randomized design with a 2 × 2 factorial arrangement of treatments: Control (animals receiving basal diet, BD without additives); FA, BD + 25 mg FA/kg; GPM, BD + 2.5% GPM/kg; and MIX, BD + 25 mg FA + 2.5% GPM/kg). The basal diet was formulated to cover the nutritional requirements for the species and productive stage [21] (Table 1). The phenolic compound content and antioxidant capacity of each experimental diet were reported in the Appendix A, respectively. 

#### 2.3.2. Productive Performance

For 10 days prior to the start of the experimental test, the pigs were fed with BD. Individual live weight was recorded at the beginning (IBW) and end (FBW) of the experimental period. Individual ADG was estimated by the difference between FBW and IBW and divided by 31 d. The FI was calculated daily, for which the weight of the feed offered and rejected per animal was recorded. Feed conversion (FC) was also estimated. All parameters were expressed in kg.

### 2.4. Environmental Conditions and Physiological Variables 

During the study, the environmental temperature and relative humidity of the herd were measured daily using a remote data logger (Hobo^®^ Model U10-003, Onset Computer Corporation, Bourne, MA, USA) in which data were recorded every 20 min. From these variables, the temperature-humidity index (THI) was calculated using the following formula: THI = (0.8 × T) + ((RH/100) × (T − 14.4)) + 46.4, and the values obtained were classified into 4 categories: suitable (<74), mild heat stress or alert zone (74 ≥ 78), moderate heat stress or danger zone (78 ≥ 82), and severe heat stress or emergency zone (≥82). These values were expressed as units THI [22].

Rectal temperature (RT) and respiratory rate (RR) of the animals were measured twice a day (0800 h and 1500 h) and three times a week [1]. The RT was measured with a calibrated digital thermometer (accuracy ± 0.2 °C). While RR was determined visually by counting the movement of the flanks for a period of 15 s [1] and the values obtained were multiplied by 4 to calculate the number of breaths per minute (bpm).

### 2.5. Slaughter and Carcass Traits

At the end of the feeding trial, all pigs were slaughtered after fasting for 16 h and this process was carried out in accordance with the regulations (NOM-033-ZOO-1995) at the slaughterhouse of the ALD of the Universidad de Sonora. The pigs were electrically stunned prior to bleeding. Live weight at slaughter, hot carcass weight, and after refrigeration at 2 °C for 24 h, cold carcass weight, rib eye area (cm^2^), and backfat thickness (mm) at the 12th rib space were measured according to [23].

### 2.6. Blood Metabolites

Blood samples were collected from 5 pigs per treatment at the beginning and end of the study. Approximately 14 mL of blood was collected by jugular venipuncture from each animal in two vacutainer tubes (Becton, Dickinson and Company, Franklin Lakes, NJ, USA), one tube with ethylenediaminetetraacetic acid (EDTA) and the other without anticoagulant and immediately placed on ice. For the hemogram test (red blood cells, hemoglobin, hematocrit, leukocytes, mean corpuscular volume (MCV), mean corpuscular hemoglobin (MCH), mean corpuscular hemoglobin concentration (MCHC), erythrocyte distribution (RDW-CV), platelets, white blood cells, granulocytes, lymphocytes, and medium cells) the blood from the EDTA-containing tubes were used. The tubes that did not contain EDTA were centrifuged at 10,000 rpm for 10 min to separate the serum from the blood cells. The serum was stored at −20 °C and the following blood parameters were evaluated: glucose, total protein, albumin, globulins, creatine kinase (CK), cortisol, growth hormone (GH), and insulin growth factor 1 (IGF-1). The parameters glucose, total protein, and albumin were quantified using the corresponding laboratory kit following the manufacturer’s instructions (RANDOX^®^ Manual). The cortisol, GH, and IGF-1 were determined using enzyme-linked immunosorbent assay (Sigma-Aldrich^®^, St. Louis, MO, USA). All analyses were performed in duplicate.

### 2.7. Statistical Analysis

Data analysis considered each animal as an experimental unit. For the analysis of variance (ANOVA) of productive performance, physiological variables, carcass characteristics, relative weight of organs, hormonal levels, and biochemical and hematological parameters a 2 × 2 factorial arrangement under a completely randomized design was used. The model considered the fixed effect of additives (FA or GPM), and their interaction (FA × GPM). Initial and final body weight were considered in the model as covariates for productive performance and carcass quality, respectively. For blood and serum parameters, the estimated initial values of the same variables were used as covariates. When significant differences were detected, the comparison of means was performed using Tukey’s test. The effect of the treatments was considered significant when *p* < 0.05 and trends were considered at *p* < 0.10. All data were analyzed with the NCSS statistical program (version 2020, NCSS, Kaysville, UT, USA).

## 3. Results

### 3.1. Quantification and Identification of Phenolic Compounds and Capacity Antioxidant of GPM

Table 2 shows the content of phenolic compounds, flavonoids, and antioxidant capacity estimated for Tempranillo grape pomace. The average content of total phenols and flavonoids in grape pomace flour was 20.81 mg GAE/g, and 11.30 mg CE/g, respectively. The antioxidant activity by FRAP, TEAC, and DPPH methods was 104.7; 139.4, and 114.8 µM TE/g, respectively.

Figure 1 shows the qualitative characterization of grape pomace (fresh, dried, and flour). From the thin layer chromatography assay, the presence of several phenolic compounds of polar and nonpolar nature at the two wavelengths evaluated together with the developed plate was verified. The presence of CGA, GA, CAT, and EC were tested and a pattern very close to the RES standard was observed (Figure 1a,b). When the plate was developed, other compounds of medium polar (retardation factors from 0.38 to 0.48) and nonpolar nature (0.66 to 0.72) were observed, different from the standards used (Figure 1c). This makes it necessary to identify and quantify by other methods the specific concentrations. Recent studies have shown that PCs such as ACG, proanthocyanidins, ellagic acid, and quercetin can promote muscle fiber transition in finishing pigs, which supports the importance of qualitatively characterizing this raw material [24].

### 3.2. Environmental Conditions and Physiological Variables

The average environment temperature was 29.9 °C, ranging from 20.41 °C to 39.06 °C, while the average relative humidity was 38.61% with a minimum and maximum of 19.8 °C and 62.28 °C, respectively (Figure 2A). In this case, the average temperature exceeded the thermal comfort zone [25] of the finishing pigs (18–25 °C). The THI in the present study ranged from 68.67 to 96.46 with an average of 76.02 units THI (Figure 2B). The THI values indicate that the animals presented a mild to severe stress level during the entire study period, therefore they were within an alert (74–78) and emergency (>82) zone, in which they must prioritize their energy reserves for thermoregulation purposes.

According to Table 3, there was no effect of any of the additives (FA or GPM) or their interaction on the physiological variables (*p* > 0.05). In the afternoon, the animals presented values of RT and RR higher than 39.3 °C and 80 bpm, respectively, which exceeded the normal values for these physiological variables and proves that the animals were under heat stress conditions during the study [1,26]. On the contrary, in the morning hours, both physiological variables remained within a normal range or thermoneutral zone [1].

### 3.3. Productive Performance of Pigs and Carcass Quality 

The effect of individual and combined supplementation of FA and GPM on the productive performance of finishing pigs is presented in Table 4. There was no significant effect of FA or GPM on FBW, ADG, and FC (*p* > 0.05). In contrast, there was an effect of the FA × GPM interaction on FI, which increased when only GPM was included in the diet, with respect to Control and diet with two additives (*p* < 0.05). Although there were no significant differences in ADG, this parameter increased by 10% in animals supplemented with GPM, and a gain of 3 kg in the FBW was obtained, in comparison to Control.

Regarding carcass characteristics (Table 5), the FA × GPM interaction tended to be significant for hot carcass yields (*p* < 0.10), while the interaction between these additives was not significant for the other variables (HCW, CCW, backfat thickness, marbling, and CCW yields). The addition of FA improved both HCW and CCW yields (*p* < 0.05). It also tended to increase HCW and CCW (*p* < 0.10). The carcass marbling degree was reduced with the addition of GPM (*p* < 0.05) and this additive tended to increase the loin area (*p* < 0.10). 

### 3.4. Relative Organ Weight of Finishing Pigs

Table 6 presents the effect of individual and combined supplementation of FA and GPM on the relative organ weight of finishing pigs (% of FP). The FA × GPM interaction was significant for the liver weight (*p* < 0.05), which decreased with the combined addition of additives, while the individual inclusion of GPM increased the percentage relative to this organ (*p* < 0.05). Likewise, the inclusion of FA tended to modify liver weight (*p* < 0.10). On the contrary, there was no significant effect of the additives on the relative weight of the spleen, heart, lung, stomach, and kidneys (*p* > 0.05).

### 3.5. Hormonal Levels, Hematological and Biochemical Parameters of Finishing Pigs

The results of hematological and biochemical parameters of pigs supplemented with FA and GPM are presented in Table 7. The values obtained were within the reference ranges reported for the species and its stage. The combined addition of FA and GPM tends to decrease MCV in finishing pigs (*p* < 0.10), while GPM exerts a significant effect (*p* < 0.05) for this variable. Likewise, the individual inclusion of GPM tended to increase erythrocyte, hemoglobin, and hematocrit values (*p* < 0.10) in contrast to FA. On the contrary, there were no significant differences in most blood constituents (MCH, MCHC, RDW-CV, platelets, white blood cells, granulocytes, lymphocytes, and medium cells), biochemical variables (glucose, total proteins, albumins, globulins, and albumin: globulin ratio and CK), and hormones (cortisol, IGF-1, and GH) (*p* > 0.05).

## 4. Discussion

The GP contains a highly variable proportion of pulp, skin, seeds, and stems, which together with the conditions of grape cultivation (ripening stage) and the production process determine its final composition. This by-product presents high variability in terms of nutritional quality and antioxidant potential. In general, grape pomace contains high amounts of soluble phenols, including flavonols (0.3–2.6 mg/g), anthocyanins (2.5 to 132 mg/g), and soluble proanthocyanins (1.2 to 68.5 mg/g) [27]. However, winemaking methods (fermentation, maceration, pressing) determine the concentration of phenolic compounds remaining in GP. Prolonged maceration times and high fermentation temperatures tend to increase the release of phenolic compounds in the wine, and the content of PCs remaining in the winemaking by-products is decreased [28,29]. In our study, the total phenolic content compounds estimated for the Tempranillo variety (20.83 mg GAE/g) agrees with the ranges reported in other studies about red GPM [16], but lower than the total phenolic content reported in grape seed meal (90.42 mg GAE/g) [30]. The GPM also exhibited considerable antioxidant capacity. Furthermore, in this by-product, we identify CGA, CA, GA, CAT, EC, and RES which exert multiple bioactivities in pigs (Figure 1) and may help to explain the effects observed in this study.

### 4.1. Physiological Variables

Rich sources in phytochemicals and especially PCs have received great interest as feed supplements to attenuate or minimize the stress to which pigs are subjected during their productive cycle [31,32]. Its beneficial effect is attributed to the antioxidant potential per se of PCs and an improvement in mitochondrial function together with an increase in antioxidant enzyme activity [33,34]. In this context, under heat stress conditions and when the critical limit of the comfort zone is exceeded, animals resort to certain mechanisms to minimize internal heat production and reach a homeostatic state [2]. Likewise, heat production and feed intake in pigs tend to decrease from 25 °C [35]. The barn temperature and relative humidity fluctuations are responsible for the increased respiratory rate and rectal temperature in finisher pigs. In this situation, pigs activate certain homeostatic mechanisms to adapt to some kind of stressor. Even, these adaptations usually have repercussions on productive performance parameters, among which feed intake has the greatest impact [2]. 

In the present study, the inclusion of the additives (FA or GPM) did not affect the RR and RT, which can be attributed to PC concentration, profile, dosage, and antioxidant capacity, among others. Likewise, other studies do not obtain changes in these variables with ferulic acid inclusion in small ruminants’ diets under heat stress conditions [36]. However, the authors demonstrated that the inclusion of FA increases the deposition of fat reserves and could modulate the animal energy metabolism at temperatures higher than the thermoneutral zone. Even, other phytochemicals such as alkaloids can partially attenuate RR and RT in growing pigs by inhibiting the Na+/K+ ATPase pump activity (thermogenesis) [32]. In addition, other PCs such as RES, quercetin, and CAT, which were identified in GPM (Figure 1) can modulate energy expenditure by inducing activity in brown adipose tissue and may act through different molecular mechanisms (SIRT1 activity, estrogen receptor stimulation, AMP-kinase signaling or sympathetic nerve activation) [37]. Although the use of natural antioxidants in monogastric diets can contribute to reducing the impact of environmental factors and increase their adaptive capacity [38] no changes were observed in our study. However, further research is needed to determine the possible mode of action of PCs as stress attenuating.

Authors should discuss the results and how they can be interpreted from the perspective of previous studies and the working hypotheses. The findings and their implications should be discussed in the broadest context possible. Future research directions may also be highlighted.

### 4.2. Productive Performance and Carcass Traits 

To obtain higher lean tissue yields in pork carcasses, multiple growth promotion technologies such as β-adrenergics (ractopamine) have been employed to increase the efficiency of dietary nutrient utilization for protein deposition and lipolytic rate [3]. However, in view of the prohibition of its use in pig production, alternatives of vegetable origin have been sought. Similarly, there are multiple pure phenolic compounds and those embedded in agro-industrial by-products that could also act as growth modulators. In this sense, a study [6] suggested that FA can act similarly to ractopamine given that it has a structure analogous to catecholamines, especially noradrenaline. With the inclusion of FA in the diets of finishing pigs, improvements have been observed in productive parameters (ADG, FC, and FBW) and carcass characteristics (loin area, and backfat thickness) [6]. In our study, it was demonstrated that the combined inclusion of FA and GPM during the finishing stage modified the FI. The inclusion of GPM (2.5%) increased FI (9%), which can be associated with the presence of aromatic compounds from this by-product that stimulate intake and make the diet more palatable [39,40]. This result is considered attractive from a production and farm programming point of view (efficient use of facilities, less time spent per animal, and higher slaughter weight in less time). Some fermentable sugars remaining in GP could be related to a higher voluntary FI, due to the pigs’ preference for sweet compounds [41]. Likewise, rich sources in PCs promote the secretion of saliva and digestive enzymes, improving nutrient absorption and utilization. Therefore, PCs have been used as feed additives to modify the organoleptic properties of monogastric diets and thus improve taste and palatability [13].

Hydroxycinnamic acids can contribute to the flavor profile through diverse mechanisms such as phenolic degradation that generates aromatic and taste compounds, imparts flavor attributes, and can modify mechanisms of the Maillard reaction [42]. However, with high doses of PCs (1500 mg/kg), FI decreased in response to a strong dietary aroma and the sensitive palate (19,000 buds) of pigs to bitter tastes [13,43]. This could explain the differences in FI observed with the combined inclusion of the two additives versus the individual inclusion of 2.5% GPM in the diet. This behavior may also be associated with a higher concentration of PCs imparting a strong aroma or some type of phenolic interaction that may be generated between the matrix and pure compounds [44]. In general, pigs show a greater response to bitter tastes and feed aversions than to preferences [42]. Furthermore, the inclusion of fibrous by-products in pigs’ diets tends to suppress FI [45,46]; however, in our study, this situation did not occur, which can be attributed to the low content of NDF, ADF (degree lignification), hydrolyzable tannins, and condensed tannins of the GPM (Table 2) [12,47].

Regarding FA supplementation, recent studies showed that the addition of FA (25 mg) in the diet of finishing pigs (Landrace × Yorkshire) improves their productive performance (FC and ADG) [6]. On the contrary, in our research, when evaluating the same dose of FA (25 mg) in crossbreeds (Yorkshire × Duroc), it did not affect the productive performance parameters. Similar results were reported by Herrera et al. [48], who evaluated low doses of FA (12–15 mg) in finishing pigs (Landrace Yorkshire Duroc), and their results were attributed to an insufficient dose to generate an effect on pigs’ performance. Likewise, multiple factors, such as purity, the vegetable origin of the FA evaluated [6,49] as well as the experimental conditions (ambient temperature and relative humidity) joined to the evaluation period, which determines nutrient utilization in pigs and may partially explain the differences between our results with Valenzuela-Grijalva study [6]. Even previous studies [48,49,50,51] have shown that breed and genetic line influence growth rate and productive performance parameters, such as feed conversion and feed intake. In this context, the Landrace and Yorkshire breeds are more efficient than the Duroc breeds in terms of ADG and FC during the finishing stage. Furthermore, this breed tends to deposit more fat than protein in the finishing stage [51]. This could explain the results reported in our study and other works [48], in which crosses with Duroc were used and no effect of FA.

Conversely, we observed that FA (25 mg) improved cold carcass and hot carcass yield, as well as tended to increase hot and cold carcass weights. These results demonstrate that pigs supplemented with FA were more efficient in utilizing dietary nutrients and depositing lean tissue. Furthermore, this helps to support and complement previous research in which lower back fat deposition (6.44 mm), increased loin area (52.45 cm^2^), higher lean efficiency (59.66%), and carcass efficiency (89.2%) were obtained with the inclusion of 15 mg and 25 mg of FA [6,48]. 

With respect to dietary inclusion of GPM, a reduction in marbling degree and a trend to increase loin area was observed. The reduction in intramuscular fat was also observed with other agro-industrial by-products (tomato, avocado meal, and citrus residues) which has been associated with its proximal composition that generates a nutrient contribution to the diet [46,52]. These results can be related to some PCs (chlorogenic acid, caffeic acid, catechin, epicatechin, and resveratrol) previously identified in GPM (Figure 1) and in other research [24,53]. Likewise, recently has been demonstrated that the inclusion of PCs such as resveratrol, and flavonoids (quercetin derivatives and anthocyanins) from mulbery in finishing pigs’ diets [24,54,55] modifies fat metabolism and reduces backfat deposition. In this sense, it has been proposed that these compounds can modulate some genes involved in fat metabolism, inhibit fatty acid synthesis, and act at the level of adiponectin receptors [24,55]. In this context, the GPM evaluated also contains compounds that could exert this modulatory effect in finishing pigs; however, future research should be conducted to corroborate this assumption and will also be confirmed by evaluating the fatty acid profile of meat supplemented with GPM or FA.

### 4.3. Relative Organ Weights of Finishing Pigs

The internal organs of monogastric can undergo multiple metabolic and structural changes in response to dietary supplementation of certain additives, phytochemicals, or growth promoters [56], which can be reflected in their relative weight and influence the final pig weight joint to its carcass yields [57]. It has been proven that secondary metabolites such as PCs could act similarly to traditional synthetic promoters (β-adrenergics) and can cross various tissues depending on their structure. In monogastric, usually, most [6] of these PCs are metabolized in the liver and intestine, where their conjugation takes place [56,58]. Therefore, previous research has evaluated the effect of multiple PCs and growth promoters on the relative weight of visceral organs involved in their metabolism which also possess specific receptors for β-adrenergics, which could help to support the results of future research. From these studies, diverse and inconsistent results have been obtained, whose effects could be attributed to the doses evaluated [58,59] and it has even been hypothesized that the size of the organs involved in PC metabolism could reflect some effect of these as a possible growth promoter [57,59]. 

In the present study, the inclusion of FA and GPM in the finishing diets modified the relative liver weight in pigs, which increased with individual supplementation of GPM and decreased with the inclusion of both additives. These results could be partially supported by the research of Njoku et al. [60] who reported an increase in relative liver weight as a function of FI and attributed this to chemical changes occurring in this organ during FI. It has also been shown that FI stimulates the growth of visceral organs, determines the distribution, and use of animal protein [61,62]. Therefore, in our research, the increase observed in the relative weight of the liver can be explained by the higher FI obtained with the GPM. In contrast, Dávila-Ramírez et al. [56] did not observe a significant effect on organ weight in finishing pigs supplemented with a commercial mixture of plant extracts (protorgan). Likewise, other studies evaluated the inclusion of 2% tannic acid in a murine model, with which no changes were obtained in the relative weight of the liver and intestine [53]. 

Regarding FA and its structural similarity to some growth-promoting compounds, Aalhus et al. [58] demonstrated that the inclusion of ractopamine hydrochloride in finishing pigs decreased liver weight but did not exert significant changes in lung and heart size. On the contrary, Bergstrom et al. [59] reported an increase in liver and heart weight as a function of ractopamine dose. Nevertheless, the authors do not explain the changes obtained in the relative weight of the liver. On the other hand, recent studies demonstrated that maternal supplementation with polyphenols exerts significant changes in the weight of the adrenal glands (not evaluated) and spleen in their progeny during the finishing stage (180 days) [63]. This information could be considered for future research with which it may be possible to prove a possible promoter effect in the adrenal glands.

### 4.4. Hematological and Biochemical Variables and Hormone Levels 

One of the most common physiological responses exhibited by animals under heat stress conditions is the decrease in hematocrit and hemoglobin content due to erythrocyte lysis and reduced erythropoiesis, which is caused by the increase in oxygen partial pressure and respiratory rate [64,65]. Likewise, under these conditions, the platelet content is modified. In this sense, previous studies have shown that the use of non-conventional feed sources rich in PCs can positively modify some hematological and biochemical parameters in monogastric under stressful situations [31,66]. Therefore, blood metabolites have been considered as indicators of nutritional metabolism and health of animals [67]. In the present study, the values obtained for hormonal levels, hematological, and biochemical parameters were within the reference ranges reported for the species and its stage [68].

However, it is shown that the joint addition of FA and GPM decreases MCV, while erythrocyte, hematocrit, and hemoglobin values tend to increase with GPM addition. Similar results were reported in other investigations [68,69] in which mixtures of herbal extracts were supplemented in finishing pig’s diets and an increase in red cell content was observed. These changes could reflect an improvement in the respiratory capacity of monogastric [70] and this information may help to support the trend presented with grape pomace on some hematological variables.

On the contrary, our experimental results differ from the study of Nicolás-López et al. [31,71], who demonstrated that the inclusion of FA in diets for finishing sheep tends to increase MCV and decrease erythrocyte content along with platelet count. In this sense, it has been proposed that FA can generate slight alterations in red cell and platelet content due to a reduction in the activity of the thyroid gland, which exerts great influence under heat stress conditions [31]. Likewise, it has been reported that this compound can increase the release of erythropoietin in the kidney, modulate erythropoiesis and exert a cytoprotective. Therefore, it is convenient to expand research in swine due to the importance of this phase in their health. Likewise, other factors such as the final live weight of the animal may influence these hematological variables since this parameter tends to affect the animal’s response to stressful situations.

Although it has been reported that PC inclusion in monogastric diets could influence the hypothalamic-pituitary-adrenal gland axis [72], no significant effect on cortisol levels was observed in our study. Similar results were reported by Dávila-Ramírez et al. [56], who also did not observe an effect of the inclusion of herbal extracts in finishing pigs. Likewise, no significant changes were observed in the levels of growth hormone and insulin-like growth factor. On the contrary, in other studies, with the inclusion of herbal extracts in growing pigs, an increase in IGF-1 levels was reported [73,74]. The differences between these results and our study can be attributed to multiple factors such as biological variability, number of animals, growth stage, dose of the compound evaluated, and experimental conditions, among others. These results demonstrate that the inclusion of GPM does not exert negative effects on blood metabolites, since they are within the reference values established for the species and its stage.

## 5. Conclusions

Dietary GPM supplementation in finishing pigs exposed to heat stress conditions can improve feed intake and some carcass traits (loin area), and modify marbling degree and relative liver weight. Although physiological variables were not modified, this source of PCs can be considered suitable as a sensory and zootechnical additive in pig production, it could also represent a promising alternative to maintain pig performance during environmental changes. While dietary FA supplementation in the finishing phase improves some carcass traits such as carcass weight and cold carcass yield, which may suggest an improvement in lean tissue deposition with similar feed intake to control. However, further research is required to clarify these results obtained with a greater number of pigs. Likewise, the results of this research contribute to expanding the information available on the use of PCs as phytogenic additives in swine production.

## Figures and Tables

**Figure 1 animals-13-02396-f001:**
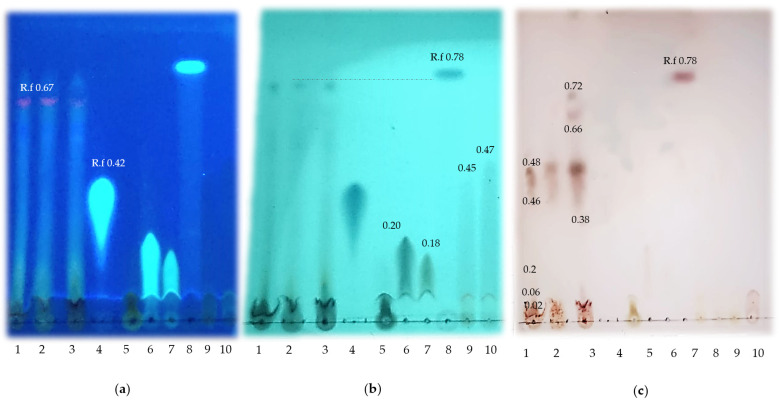
Qualitative characterization of GPM by thin layer chromatography. (**a**) UV (254 nm); (**b**) UV (365 nm); (**c**) Revealed plate. 1 (sample GPM), 2 (Sample dried GP); 3 (sample fresh GP); 4 (ferulic acid); 5 (GA); 6 (CGA), 7 (CA); 8 (RES); 9(CAT); 10 (EC). R.f (Retard factor).

**Figure 2 animals-13-02396-f002:**
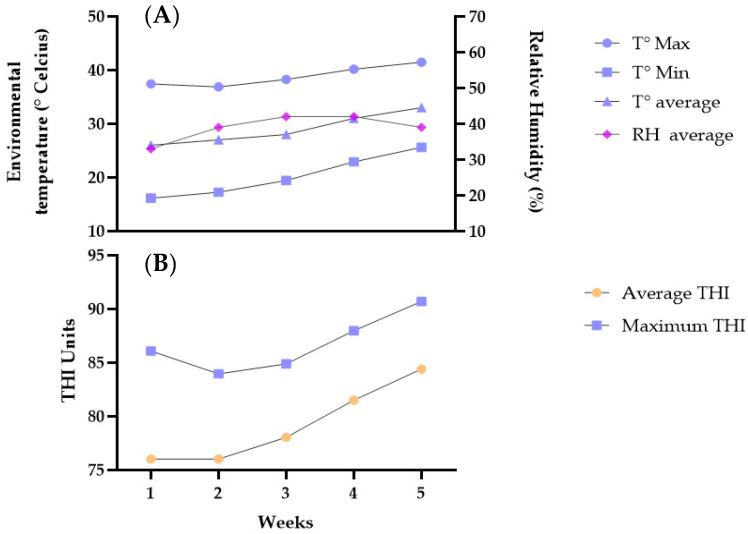
Environmental temperature and relative humidity: Maximum (T° Max), minimum (T° Min), and average temperature (T° average); Relative humidity average (RH average) (**A**). Temperature-Humidity index units (THI), Average THI and maximum THI (**B**).

**Table 1 animals-13-02396-t001:** Ingredients and chemical composition of experimental diets.

Ingredients	Treatments ^1^			
	Control	FA	GPM	MIX
Wheat grain, %	76.2	76.2	73.7	73.7
Soybean meal, %	17.0	17.0	17.0	17.0
Vegetable oil, %	4.4	4.4	4.4	4.4
Premix ^2^, %	2.4	2.4	2.4	2.4
GPM, %	0.0	0.0	2.5	2.5
FA, mg/kg	--	25	--	25
Proximate Composition				
Crude protein, %	14.0	14.0	13.9	13.9
Moisture, %	11.9	11.9	11.9	11.9
Fat, %	7.0	7.0	7.0	7.0
Fiber crude, %	2.0	2.0	2.1	2.1
Ash, %	7.0	7.0	7.0	7.0
NFE ^3^, %	58.1	58.1	58.1	58.1
ME ^4^, Mcal/kg	3.35	3.35	3.34	3.34

^1^ Control: animals receiving basal diet, DB without additives; FA: DB + 25 mg FA/kg; GPM: BD + 2.5% GPM/kg; and MIX: BD + 25 mg FA + 2.5% GPM/kg). ^2^ Premix: Premix of amino acids, vitamins, and minerals. Each kilogram of feed provided 9.5 g dicalcium phosphate, 8.3 g limestone, 3.55 g sodium chloride, 2.3 g L-lysine, 0.5 g DL-methionine, 0.35 g L-threonine, 0.15 g L-tryptophan, 80 mg DL-tocopherol acetate, 2.2 g retinol acetate, 16.5 mg cholecalciferol, 4.4 mg sodium bisulfite, 242 mg choline, 33 mg niacin, 8.8 mg riboflavin, 24.2 mg D-pantothenic acid, and 0.04 mg vitamin B12. ^3^ NFE: Nitrogen-free extract. ^4^ ME: Metabolizable energy, mega-calories per kg.

**Table 2 animals-13-02396-t002:** Phenolic content and antioxidant capacity of GPM.

Variable	Value
Total phenols compounds, mg GAE/g	20.81
Flavonoids, mg/CE/g	11.30
Hydrolysable Tannins, mg GAE/g	3.34
Condensed Tannins, mg/CE/g	0.8
Anthocyanins, mg/CE/g	1.08
FRAP, µM TE/g	104.7
TEAC, µM TE/g	139.4
DPPH, µM TE/g	114.8

*n* = 3, mg GAE/g (mg gallic acid equivalents/g); mg CE/g (mg catechin equivalents/g); µM TE/g (µmol Trolox equivalent/g); FRAP (Ferric reducing antioxidant power); TEAC (Trolox equivalent capacity); DDPH (2,2-diphenyl-1-picrylhydrazyl).

**Table 3 animals-13-02396-t003:** Rectal temperature and respiratory rate of finishing pigs supplemented with ferulic acid and grape pomace meal.

		Treatments					*p*-Values		
		FA, mg		GPM, %			FA	GPM	FA × GPM
		0	25	0	2.5	SEM			
AM	RT, °C	38.65	38.7	38.69	38.66	0.004	0.521	0.611	0.931
	RR, bpm	51.9	50.59	53.26	49.23	2.30	0.694	0.232	0.963
PM	RT, °C	39.39	39.46	39.4	39.45	0.005	0.361	0.551	0.162
	RR, bpm	82.55	82.21	83.23	81.51	3.57	0.673	0.496	0.622

FA: ferulic acid; GPM: grape pomace meal; RT: rectal temperature; RR: respiratory rate. SEM: standard error of the mean.

**Table 4 animals-13-02396-t004:** Productive performance of finishing pigs supplemented with ferulic acid and grape pomace meal.

Variable	Treatments						*p*-Value		
	FA	0		25 mg			FA	GPM	FA × GPM
	GPM	0	2.5%	0	2.5%	SEM			
IBW, kg		79.37	80.86	81.06	79.62	2.92	0.931	0.991	0.624
FBW, kg		116.03	119.09	116.28	116.62	1.38	0.432	0.331	0.231
ADG, kg		1.15	1.27	1.17	1.16	0.04	0.273	0.224	0.144
FI, kg DM/d		2.74 a	2.99 b	2.86 ab	2.76 a	0.08	0.461	0.344	0.038
FC, kg DM		2.39	2.38	2.40	2.39	0.07	0.552	0.617	0.692

FA: ferulic acid; GPM: grape pomace meal; IBW: initial body weight; FBW: final body weight; ADG: average daily gain; FI: feed intake; FC: feed conversion; DM, dry matter. Means with different letters indicate significant differences (*p* < 0.05). SEM: standard error of the mean.

**Table 5 animals-13-02396-t005:** Carcass traits of pigs supplemented with ferulic acid and grape pomace meal.

Variables	Treatments						*p*-Value		
	FA	0		25 mg			FA	GPM	FA × GPM
	GPM	0	2.5%	0	2.5%	SEM			
HCW, kg		87	86.06	87.11	88.59	0.75	0.088 *	0.728	0.123
CCW, kg		85.85	84.68	86.17	87.25	0.74	0.063 *	0.956	0.152
HCW yields, %		82.41	81.21	82.62	83.61	0.53	0.022	0.851	0.056*
CCW yields, %		81.56	80.39	82.1	82.72	0.58	0.020	0.652	0.144
pH_24_		5.52	5.51	5.5	5.5	0.04	0.741	0.913	0.841
Backfat thickness, mm		10.41	10.07	11.07	10.17	1.53	0.801	0.692	0.851
Marbling score		2.75	2.44	3.017	2.35	0.23	0.723	0.049	0.472
Loin area, cm^2^		57.23	59.77	59.49	61.4	1.25	0.13	0.095 *	0.822

FA: ferulic acid; GPM: grape pomace meal; HCW: hot carcass weight; CCW: cold carcass weight. * Trends (*p* < 0.10). SEM: standard error of the mean.

**Table 6 animals-13-02396-t006:** The relative weight of the organs of finisher pigs supplemented with grape pomace and ferulic acid.

Relative Weight Organ	Treatments						*p*-Value		
	FA	0		25 mg			FA	GPM	FA × GPM
	GPM	0	2.5%	0	2.5%	SEM			
Liver, %		1.59 ab	1.69 a	1.60 ab	1.50 b	0.04	0.078 *	0.982	0.040
Spleen, %		0.17	0.18	0.18	0.17	0.01	0.867	0.831	0.279
Heart, %		0.36	0.36	0.35	0.36	0.02	0.744	0.803	0.983
Lung, %		1.02	0.99	0.97	0.87	0.05	0.106	0.225	0.537
Stomach, %		0.52	0.57	0.52	0.50	0.02	0.138	0.657	0.138
Kidney, %		0.34	0.34	0.37	0.33	0.01	0.458	0.233	0.212

FA: ferulic acid; GPM: grape pomace meal; Means with different letters indicate significant differences (*p* < 0.05). * Trends (*p* < 0.10). SEM: standard error of the mean.

**Table 7 animals-13-02396-t007:** Hematological and biochemical parameters of finishing pigs supplemented with ferulic acid and grape pomace.

Variables	Treatments						*p*-Value		
	FA	0		25 mg		SEM	FA	GPM	FA × GPM
	GPM	0	2.5%	0	2.5%				
Hematological parameters									
Red blood cells (10^12^/µL)		5.16	5.22	4.92	5.44	0.14	0.931	0.062 *	0.137
Hemoglobin, g/dL		15.24	15.40	14.54	15.91	0.40	0.810	0.082 *	0.169
Hematocrit, %		45.08	45.51	42.91	46.98	1.19	0.770	0.081 *	0.158
MCV, fL		87.1 ab	87.09 ab	87.28 a	86.33 b	0.19	0.170	0.031	0.050
MCH, pg		29.55	28.34	29.47	29.56	0.61	0.533	0.268	0.469
MCHC, g/dL		33.81	33.87	33.87	33.84	0.02	0.200	0.882	0.331
RDW-CV, %		18.07	18.61	19.66	18.5	0.64	0.273	0.665	0.256
Platelets (10^3^/µL)		218.0	220.3	163.8	224.6	25.5	0.349	0.245	0.275
White blood cells (10^9^/µL)		15.32	17.44	17.039	17.236	1.32	0.584	0.409	0.483
Granulocytes, %		35.39	33.45	39.51	34.33	2.77	0.384	0.259	0.577
Lymphocytes, %		52.13	56.03	50.34	54.40	2.53	0.512	0.184	0.975
Medium cells, %		11.1	9.99	11.02	10.44	1.53	0.905	0.609	0.865
Biochemical parameters									
Glucose, mg/dL		66.55	69.035	69.41	76.76	5.16	0.290	0.345	0.729
Total proteins, g/dL		6.475	6.493	6.524	6.347	0.22	0.839	0.733	0.674
Albumin, g/dL		4.163	3.457	3.147	3.416	0.30	0.221	0.264	0.268
Globulins, g/dL		2.436	3.008	3.064	2.867	0.30	0.440	0.550	0.220
A: G ratio		2.3	1.276	1.243	1.166	0.51	0.281	0.312	0.379
CK, U/L		1290.5	1695.6	1141.6	1430.9	226.3	0.386	0.148	0.813
Hormonal levels									
Cortisol, µg/dL		2.189	1.563	2.275	1.66	0.39	0.82	0.155	0.982
GH, ng/mL		0.143	0.086	0.036	0.456	0.20	0.572	0.400	0.309
IGF-1, ng/mL		158.37	102.02	132.05	155.73	26.61	0.634	0.551	0.164

FA: ferulic acid; GPM: grape pomace meal. MCV: mean corpuscular volume; MCH: mean corpuscular hemoglobin; MCHC: mean corpuscular hemoglobin concentration; RDW-CV: erythrocyte distribution; CK: creatine kinase; GH: growth hormone; IGF-1: insulin growth factor 1. A: G, albumin: globulins ratio. Means with different letters indicate a significant difference (*p* < 0.05). * Trends (*p* < 0.10). SEM: Standard error of the mean.

## Data Availability

The data sets analyzed in the present study are available from the corresponding authors upon request.

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
