# Peer review of "Productive Performance, Physiological Variables, and Carcass Quality of Finishing Pigs Supplemented with Ferulic Acid and Grape Pomace under Heat Stress Conditions"

_animals, 2023, doi:10.3390/ani13142396_

Round 1

Reviewer 1 Report (Previous Reviewer 3)

The authors have addressed my previous comments. 

Some specific comments 

Line 50: feedlot performance is a strange term to use for pigs. Consider changing feedlot to “production” or simply “growth”

Line 140 – 143: the new section added is quite confusing and needs to be rewritten for clarity. It is not necessary to include “uniformity in” and the parentheses can be removed around the weights.

The animals were individually housed (0.6 x 2.0 m) in open buildings representing local commercial facilities for pigs during the grow – finish phase. Pigs were given ad libitum access to nipple-type drinkers and feeding troughs

Line 156: change “final” to “end”

There are some language issues throughout the text to be corrected prior to publication. 

In many cases, there wrong tense is used and the use of words like on/in/of needs to be checked 

Author Response

Reviewer 1

The authors have addressed my previous comments. 

 R: The authors appreciate their comments and observations, which were considered in the corrected version of the manuscript. Changes were indicated in blue text.

Some specific comments 

Line 50: feedlot performance is a strange term to use for pigs. Consider changing feedlot to “production” or simply “growth”.

R: Thanks. Feedlot peormance was changed to “growth perfomance”. Line 50.

Line 140 – 143: the new section added is quite confusing and needs to be rewritten for clarity. It is not necessary to include “uniformity in” and the parentheses can be removed around the weights.

The animals were individually housed (0.6 x 2.0 m) in open buildings representing local commercial facilities for pigs during the grow – finish phase. Pigs were given ad libitum access to nipple-type drinkers and feeding troughs.

R: Thanks for this recommendation. The paragraph was rewritten. Now: Forty male pigs from commercial Duroc x Yorkshire cross breed with an initial live weight of 80.2±4.6 kg were used. The animals were individually housed (0.6 x 2.0 m) in open buildings representing local commercial facilities for pigs during the grow-finish phase. Pigs were given ad libitum access to nipple-type drinkers and feeding troughs. Lines 141-145.

Line 156: change “final” to “end”

R: Attended. Line 157

Comments on the Quality of English Language

There are some language issues throughout the text to be corrected prior to publication. 

In many cases, there wrong tense is used and the use of words like on/in/of needs to be checked 

R: An English language grammar check of this version of the manuscript was performed.

Reviewer 2 Report (New Reviewer)

The paper entitled "Productive Performance, Physiological Variables, and Carcass Quality of Finishing Pigs Supplemented with Ferulic Acid and Grape Pomace under Heat Stress Conditions" is interesting. 

Firstly, the paper presents a comprehensive and well-designed study that investigates the effects of ferulic acid and grape pomace supplementation on the productive performance, physiological variables, and carcass quality of finishing pigs under heat stress conditions. The study was carefully conducted, with appropriate  statistical analyses, ensuring that the results are reliable and valid, however the sample sizes should be more precisly described. 

Secondly, the study addresses an important issue in animal agriculture, namely the impact of heat stress on pig production and the potential benefits of dietary interventions in minimizing these effects. The study contributes to the existing body of knowledge on this topic, providing valuable insights into how ferulic acid and grape pomace may improve the performance and health of pigs subjected to heat stress.

Thirdly, the paper is well-written, clearly structured, and follows the guidelines established by the journal. The authors have provided a detailed description of their methods, results, and conclusions, making it easy for readers to understand and interpret the findings.

Overall, the paper entitled "Productive Performance, Physiological Variables, and Carcass Quality of Finishing Pigs Supplemented with Ferulic Acid and Grape Pomace under Heat Stress Conditions" represents a high-quality research study that addresses an important issue in animal agriculture. 

Minor corrections: Table 1, give the total 100% of the ingredients. 

Please provide how many animals were included in the study and how many were slaughtered. Also please mention if some analyses were determined in duplicate. 

It will be interesting if the authors will explain why there is such difference in the antioxidant capacity of grape pomace meal and grape seed meal, as some studies reported higher values than the authors in this study, for example TPC in grape seed meal was 90.42 versus 20.81 in the pomace (https://doi.org/10.1038/s41598-021-00343-1)

Exclude the bold from the text when indicating the tables. 

Use suprscripts letter when indicating the significance among the groups in the tables. 

Best of luck!

Author Response

Reviewer 2
The paper entitled "Productive Performance, Physiological Variables, and Carcass Quality of Finishing Pigs Supplemented with Ferulic Acid and Grape Pomace under Heat Stress Conditions" is interesting. 

Firstly, the paper presents a comprehensive and well-designed study that investigates the effects of ferulic acid and grape pomace supplementation on the productive performance, physiological variables, and carcass quality of finishing pigs under heat stress conditions. The study was carefully conducted, with appropriate  statistical analyses, ensuring that the results are reliable and valid, however the sample sizes should be more precisly described. 

Secondly, the study addresses an important issue in animal agriculture, namely the impact of heat stress on pig production and the potential benefits of dietary interventions in minimizing these effects. The study contributes to the existing body of knowledge on this topic, providing valuable insights into how ferulic acid and grape pomace may improve the performance and health of pigs subjected to heat stress.

Thirdly, the paper is well-written, clearly structured, and follows the guidelines established by the journal. The authors have provided a detailed description of their methods, results, and conclusions, making it easy for readers to understand and interpret the findings.

Overall, the paper entitled "Productive Performance, Physiological Variables, and Carcass Quality of Finishing Pigs Supplemented with Ferulic Acid and Grape Pomace under Heat Stress Conditions" represents a high-quality research study that addresses an important issue in animal agriculture. 

R: The authors appreciate your comments and opinions on the manuscript. Changes were indicated in blue text.

Minor corrections:

Table 1, give the total 100% of the ingredients. 

R: All ingredients of the experimental ration were reported on Table 1.

Please provide how many animals were included in the study and how many were slaughtered. Also please mention if some analyses were determined in duplicate. 

R: The number and characteristics of the animals used in the study were indicated in Lines 141-142. All pigs were slaughtered and was indicated on line 188. Some analyzes were performed in duplicate and others in triplicate, and it was also indicated in the manuscript (Lines 96, 118, 211).

It will be interesting if the authors will explain why there is such difference in the antioxidant capacity of grape pomace meal and grape seed meal, as some studies reported higher values than the authors in this study, for example TPC in grape seed meal was 90.42 versus 20.81 in the pomace (https://doi.org/10.1038/s41598-021-00343-1).

R: Thanks for your recommendation.

The following reference was included to the manuscript, and TPC values of GPM were compared with those from seed grape meal (lines 386-387):

Vlaicu, P.A., Panaite, T.D. & Turcu, R.P. Enriching laying hens eggs by feeding diets with different fatty acid composition and antioxidants. Sci Rep 11, 20707 (2021). https://doi.org/10.1038/s41598-021-00343-1.

Exclude the bold from the text when indicating the tables. 

R: Attended

Use superscripts letter when indicating the significance among the groups in the tables. 

R: Attended

This manuscript is a resubmission of an earlier submission. The following is a list of the peer review reports and author responses from that submission.

Round 1

Reviewer 1 Report

The main theme of this study is Productive performance, physiological variables, and carcass quality of finishing pigs supplemented with ferulic acid and grape pomace subjected to heat stress conditions. However, the objectives, experimental design and methods used in this study are inadequate and innovative, and the experimental design is too simple. Suggested simplification of the description of the data analysis section. In addition, the presentation of data in the text is rather homogeneous, being only in tables, and it is recommended to diversify and improve the quality of the article.

/

Reviewer 2 Report

Dear authors,

 Firstly, I highly appreciate the authors’ topic. Ospina-Romero et al. tested to determine the productive performance, physiological variables, and carcass quality of finishing pigs supplemented with ferulic acid and grape pomace subjected to heat stress conditions.

It is a meaningful study for scientists, farmers, and animal feed companies. The parts of the manuscript are written well. However, some parts should be logically changed and the following questions should be clearly explained:

1)     Table 1 (lines 124-127) should be moved to result place

2)     “Northwestern Mexico” should be added after Hermosillo (line 147)

3)     Daily temperature and humidity during the entire experiment should be presented in a table because it is essential for changing studied variables.

4)     ME should be supplemented in Table 2

5)     In part 2.3.1, the authors should describe the detail of the experiment as:

Where are 40 male pigs from? From how many sows, litters,…

Pigsty of the experiment is open or closed?

Each pen has how surface?

6)     Supplementary Table S1 (line 289) should be presented in the detailed table

7)     What is RF (line 328)?  

Therefore, in my opinion, the revised manuscript is not enough strong to persuade for reviewing. It needs revised and add necessary information to persuade for reviewing and to meet Animals standards.

Reviewer 3 Report

Please see attached document with my comments 

Some corrections should be made - tenses used, plurals of words etc. 

In addition, please check the length of the sentences in introduction and discussion especially. Many are too long and need to be split in multiple or shortened